# Molecular identification and quantification of defect sites in metal-organic frameworks with NMR probe molecules

Jinglin Yin[1,2,5], Zhengzhong Kang[1,5], Yao Fu[1], Weicheng Cao[1], Yiran Wang[1], Hanxi Guan[1], Yu Yin[1], Binbin Chen[1], Xianfeng Yi [3], Wei Chen [3], Wei Shao[4], Yihan Zhu [4], Anmin Zheng [3], Qi Wang [1] & Xueqian Kong [1,2]✉

The defects in metal-organic frameworks (MOFs) can dramatically alter their pore structure and chemical properties. However, it has been a great challenge to characterize the molecular structure of defects, especially when the defects are distributed irregularly in the lattice. In this work, we applied a characterization strategy based on solid-state nuclear magnetic resonance (NMR) to assess the chemistry of defects. This strategy takes advantage of the coordination-sensitive phosphorus probe molecules, e.g., trimethylphosphine (TMP) and trimethylphosphine oxide (TMPO), that can distinguish the subtle differences in the acidity of defects. A variety of local chemical environments have been identified in defective and ideal MOF lattices. The geometric dimension of defects can also be evaluated by using the homologs of probe molecules with different sizes. In addition, our method provides a reliable way to quantify the density of defect sites, which comes together with the molecular details of local pore environments. The comprehensive solid-state NMR strategy can be of great value for a better understanding of MOF structures and for guiding the design of MOFs with desired catalytic or adsorption properties.

Metal-organic frameworks (MOFs) are an emerging class of porous materials that have enormous potentials in gas separation[1-4], catalysis[5-10], drug delivery[11-15], and energy storage[16-19]. Typically, MOFs are viewed as crystalline lattices with accessible regular-shaped pores. However, the prevalence of defects in MOFs as discovered in recent works has challenged the conventional perception[20-22]. The defects could fundamentally alter the chemical environment[23-25] as well as the spatial geometry and connectivity of the pore space[26-29]. It has been suggested by various studies that the defects could be one of the key contributors to the catalytic activities[30-34], and could significantly affect adsorption and transport properties[29,35-39]. The characterizations of defects are technically challenging as the structure of defect sites may be disordered and heterogeneous, and they could locate internally or at the particle surface[40,41]. Thermogravimetric analysis[42,43] and other measurements[44-47] can only provide limited site-specific information of defects. The recent development of low dose high-resolution transmission electron microscopy (HRTEM) or cryogenic electron microscopy (cryo-EM) has made great progress to directly visualize defect formations[48-53]. Yet the imaging techniques, at the current stage, cannot identify irregularly-distributed defects or

[1]Department of Chemistry, Zhejiang University, 310027 Hangzhou, P. R. China. [2]Key Laboratory of Excited-State Materials of Zhejiang Province, Zhejiang University, 310027 Hangzhou, P. R. China. [3]State Key Laboratory of Magnetic Resonance and Atomic and Molecular Physics, National Center for Magnetic Resonance in Wuhan, Wuhan Institute of Physics and Mathematics, Innovation Academy for Precision Measurement Science and Technology, Chinese Academy of Sciences, 430071 Wuhan, P. R. China. [4]College of Chemical Engineering and State Key Laboratory Breeding Base of Green Chemistry Synthesis Technology, Zhejiang University of Technology, 310014 Hangzhou, China. [5]These authors contributed equally: Jinglin Yin, Zhengzhong Kang. ✉e-mail: kxq@zju.edu.cn

provide quantifications of bulk phases. In addition, the chemical functionalities (e.g., hydroxyl, water, and small-molecule modulators) and the acid-base properties are largely absent in the pictures of defects.

Solid-state nuclear magnetic resonance (SSNMR) can thoroughly elucidate the chemical characters of organic moieties and metal centers in the defective framework[54–59]. However, the direct characterizations of defects are limited by resolution and sensitivity, and lack of spatial and volumetric information.[25,39,60,61] In this work, we present an indirect strategy that utilizes phosphorus-containing probe molecules to characterize defects in MOFs, which can provide both molecular details and bulk quantifications. The indirect method is more sensitive and background-free (based on the receptive [31]P signal), and is highly chemical-specific and quantitative. By varying the size of probe molecules (i.e., by controlling their accessibility to different pores), we can also distinguish between defects of different geometric volumes[62] as well as their distributions. Phosphorus-containing probe molecules (e.g., the homologs of trimethylphosphine oxide, TMPO, and trimethylphosphine, TMP) have been adopted to study the acidity of solid-acid catalysts and zeolites[63–67]. The translation of these methods into MOFs, particularly targeting defects, requires a comprehensive understanding of the host-guest interactions between the probe molecules and the MOF matrices. Here, we combined [31]P NMR methods and density functional theory (DFT) calculations to identify the one-to-one correspondence between the [31]P signals and the underlying defect structures. The defect sites terminated with or without modulator can be distinguished in [31]P spectra from that of the ideal framework. With the molecular understanding of the defect-associated signals, the quantification of defects in MOFs was achieved. It's expected that our NMR strategy can provide much-needed chemical resolution and quantitative accuracy to the frontier explorations of defects in MOFs and possibly in zeolitic imidazolate frameworks (ZIFs)[68,69] and covalent organic frameworks (COFs)[70,71] as well.

## Results and discussion
### The loading of probe molecules
In this work, we focus on the MOF named UiO-66(Zr), as its defects are one of the most researched in the literature. The defective UiO-66 samples are prepared according to the reported procedures[25] with or without acetic acid as the modulator (labeled as "dU"). The "ideal" UiO-66 sample is prepared by using HCl to control the pH of solution[42] (labeled as "iU"). The formation of defects was determined by the thermogravimetric analysis (TGA) (see the descriptions in the Methods section). Representative transmission electron microscopy (TEM) images of MOF particles are shown in Fig. 1a. In general, the defective UiO-66 appears as separated particles while the ideal ones are aggregated (Supplementary Fig. 1). The powder x-ray diffraction (PXRD) of

these samples show expected patterns as the simulated structure (Fig. 1b). The low angle region ($2\theta = 3°–7°$) of our dU samples is largely flat, suggesting the distribution of defects is not as ordered as those in defective UiO-66(Hf) modulated with formic acid[53,72].

TMP, TMPO, tributylphosphine oxide (TBPO), and trioctylphosphine oxide (TOPO) (Fig. 1c) are used as the probe molecules in our NMR investigations. The loading of probe molecules is performed on the samples activated under vacuum at 150 °C. The adsorption procedure includes adsorption, equilibration, and solvent removal steps (see Methods section for details). Note that TMP is easily oxidized into TMPO, while TMPO, TBPO, and TOPO are sensitive to moisture. The samples were prepared in Ar or $N_2$ glove boxes to avoid the environmental influence. PXRD patterns of the loaded samples are recorded to ensure that the crystalline structure of UiO-66 is preserved (Supplementary Fig. 2). The nitrogen sorption data were also collected on samples before and after TMPO/TMP adsorption (Supplementary Fig. 3). The probe-loaded samples are packed in solid-state NMR rotors and their [31]P spectra are recorded under magic angle spinning (MAS). The detailed experimental procedures and characterization results can be found in the Methods section.

### Identify the local acidity of defects
Both Lewis and Brønsted acid sites can be present in MOFs. The μ-OH that exists in both perfect lattice and defective frameworks are Brønsted acid sites. In a defective MOF, some linkers are missing on defect zirconium atoms. Normally, most of the defect zirconium are compensated by acetate modulators and water (in dU-Ac) creating new Brønsted acid sites. Only a few modulators could fall off, leaving under-coordinated zirconium as Lewis acid sites. In contrast, the under-coordinated zirconium sites could be abundant in structures lack of acetate (in dU-no Ac).

The [31]P chemical shift ($\delta_{31P}$) of TMP can be a sensitive probe to distinguish Lewis and Brønsted sites. It has been well researched that, for adsorbed TMP, the [31]P signals in the range of −60 to −63 ppm correspond to physical adsorption, those in the range of −25 to −60 ppm correspond to Lewis acid sites, and those in the range of −2 to −5 ppm correspond to Brønsted acid sites[66,73]. The TMP in ideal UiO-66 has only one signal at −63 ppm indicating only physically adsorbed molecules (Fig. 2a). The TMP does not bond to the μ-OH in ideal UiO-66 suggesting the Brønsted acidity of μ-OH in ideal UiO-66 is relatively weak.

In the defective UiO-66 with acetate modulator (dU-Ac), Brønsted acid sites ($\delta_{31P} = −2$ to −3 ppm) appear. According to the [1]H-[31]P two-dimensional (2D) heteronuclear correlation (HETCOR) spectrum (Fig. 2b), we attribute the Brønsted sites to the defects bonded with water ($\delta_{1H} = 4.5$ ppm). In the defective UiO-66 without acetate modulator (dU-no Ac), we can observe the presence of Lewis acid sites

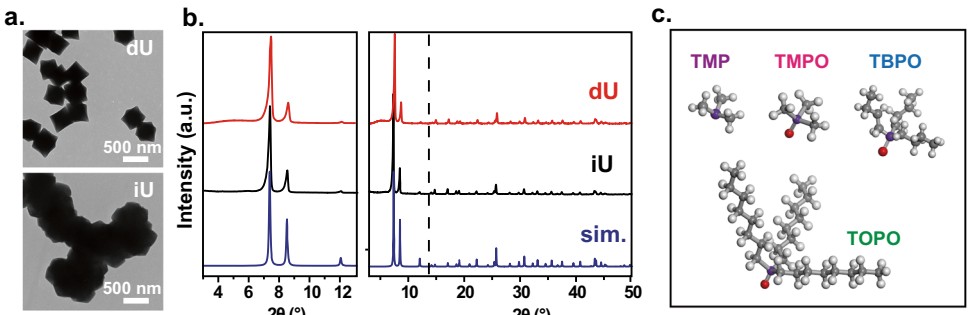

**Fig. 1 | Standard characterizations. a** TEM images of UiO-66 particles. **b** The PXRD patterns of ideal and defective UiO-66. **c** The chemical structures of [31]P probe molecules used in this work. "dU" represents defective UiO-66, "iU" represents ideal UiO-66, and "sim." corresponds to the PXRD simulation result. TMP:

trimethylphosphine, TMPO: trimethylphosphine oxide, TBPO: tributylphosphine oxide, TOPO: trioctylphosphine oxide. The atoms are colored red for O, gray for C, purple for P, and white for H.

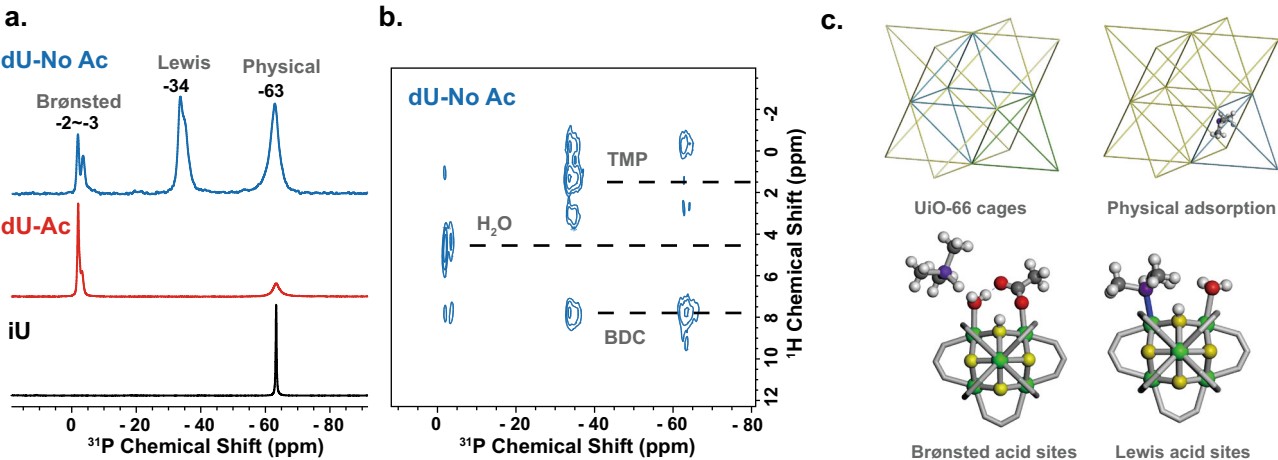

**Fig. 2 | Adsorption of TMP. a** $^1$H-$^{31}$P cross-polarization (CP) spectra of TMP adsorbed in dU-No Ac (defective UiO-66 without acetate), dU-Ac (defective UiO-66 with acetate) and iU(ideal UiO-66). The CP contact time is 2 ms. The signals for physical adsoption, Lewis acid sites, and Brønsted acid sites are marked

respectively. **b** $^1$H-$^{31}$P 2D HETCOR spectrum of TMP loaded in dU-No Ac. **c** Schematic illustration of UiO-66 cages and the configurations of adsorbed TMP. The atoms are colored green for Zr, yellow for μ-O, red for O at defect sites, gray for C, purple for P, and white for H.

($\delta_{31P}$ = −34 ppm). The Lewis sites are attributed to under-coordinated zirconium atoms which are exposed when there is no acetate modulator. When TMP is coordinated to the Lewis acid sites, the $^1$H signal of its methyl groups appears as several peaks ($\delta_{1H}$ = −0.5–3 ppm) which could be due to the differences in pore environment.

The assignment of TMP signals is further supported by DFT calculations of $^{31}$P chemical shift. Figure 2c presents the probable structures for DFT calculation including the physical adsorbed TMP in ideal UiO-66 cages, and TMP at the Brønsted and Lewis defect sites. The calculated shifts (listed in Supplementary Table 1) are in good agreement with experimental results. The calculation shows that the TMP coordinated to the Brønsted site is protonated (i.e., TMPH$^+$). Here, the Brønsted site is formed by the coexistence of water and acetate.[25] As UiO-66 was prepared in acidic environment, there are extra protons that may be captured by TMP to form TMPH$^+$. The TMP coordinated to Lewis site may also be accompanied by water or acetate as the neighbor, yet the hydroxyl is improbable due to the unfavorable energetics (Supplementary Fig. 4). The physically adsorbed TMP is likely trapped at the window between the small tetrahedral cage and the large octahedral cage[74,75].

Using TMP as the probe, we are able to distinguish the defects with open zirconium sites (Lewis site) from those occupied by water/ acetate (Brønsted site). However, in UiO-66 prepared with acetate modulator, Brønsted defect sites dominate. Yet the $^{31}$P shift of TMP is not very sensitive to Brønsted acidity, as compared to TMPO. It is because that TMPO has an electron-withdrawing oxygen atom (prefers the H$^+$ of Brønsted sites), while TMP has an electron-donating lone pair (prefers the electron-deficient Lewis sites). In the next section, we use TMPO to reveal the fine structures of Brønsted defect sites.

### The fine structure of Brønsted defects

The $^{31}$P spectra of TMPO-loaded UiO-66 samples (Fig. 3a, b) show at least eight different sites (labeled from A to H). In general, these species can be categorized into physically adsorbed species (from 30 to 50 ppm) and chemically adsorbed species (from 50 to 70 ppm)[76]. Most physically adsorbed and some chemically adsorbed species can be washed away by dichloromethane, yet some chemically adsorbed species cannot be washed away. This suggests the variable bonding strengths and configurations of TMPO in UiO-66. The accurate assignment of these species is challenging and requires a collaborative effort of NMR and DFT calculations.

Let us focus on the physically adsorbed species first. The $^{31}$P signals at 32 and 36 ppm (species H and G, respectively) are shifted

upfield from that of pure crystalline TMPO ($\delta_{31P}$ = 42 ppm). This is an indication of the shielding effect of benzene dicarboxylate (BDC) linkers. Calculations show the H and G species are TMPO being attracted inside or towards the small tetrahedral cages. 2D $^{31}$P-$^{31}$P correlation spectrum (Fig. 3c) shows the cross peaks between the H and G species indicating these two species are in spatial proximity, i.e., in the same or neighboring small cages. The species E ($\delta_{31P}$ = 46 ppm) and F ($\delta_{31P}$ = 42 ppm) can be attributed to physically adsorbed molecules in the large cage. According to the DFT calculation, the F species is the isolated TMPO while the E species is hydrogen-bonded with water (Fig. 3d). The assignment is corroborated by the intentionally water-treated sample which shows a significant enhancement of E signal (Supplementary Fig. 5).

For chemically adsorbed TMPO, we attribute the signals greater than 60 ppm to the species bonded to Lewis acid sites and the signals between 60 to 50 ppm to those bonded to Brønsted acid sites, based on our DFT calculations (Supplementary Table 2). The A species ($\delta_{31P}$ = ~62 ppm) only weakly appears in the dU-Ac sample. Since most Lewis sites in dU-Ac are occupied by acetate and water, the weak signal of A is well expected. Both defective and ideal UiO-66 have a strong signal of B species ($\delta_{31P}$ = ~58 ppm). The B species can be attributed to TMPO bonded to the μ-OH in non-defective cages (Fig. 3d). The species C ($\delta_{31P}$ = ~55 ppm) and D ($\delta_{31P}$ = ~53 ppm) only show up in defective UiO-66. Species C is attributed to TMPO bonded to the μ-OH next to a missing BDC linker, while species D can be attributed to TMPO bonded to the defect-associated water. The combinative use of TMP and TMPO probe molecules can provide a holistic and detailed picture of the internal chemistry in MOFs.

### The volumetric accessibility of defects

Besides the chemical properties of defects, the accessibility of defects, based on their pore volume and connectivity, is also important. The accessibility of active sites can have a direct consequence on the catalytic activity, biological functionality, and adsorption properties[39,77,78]. The evaluation of accessibility can be achieved by varying the size of probe molecules. Here, we use the homologs of TMPO, i.e., TMPO, TBPO, and TOPO which have kinetic diameters of 0.55, 0.82, and 1.1 nm, respectively[79].

In ideal UiO-66, the windows between cages are about 0.5–0.7 nm[80], which allows the uptake of only TMPO. While in defective UiO-66, the windows open wider and could allow the adsorption of bigger molecules. Therefore, TMPO can access both ideal and defective internal space of UiO-66, while TBPO and TOPO can only access a

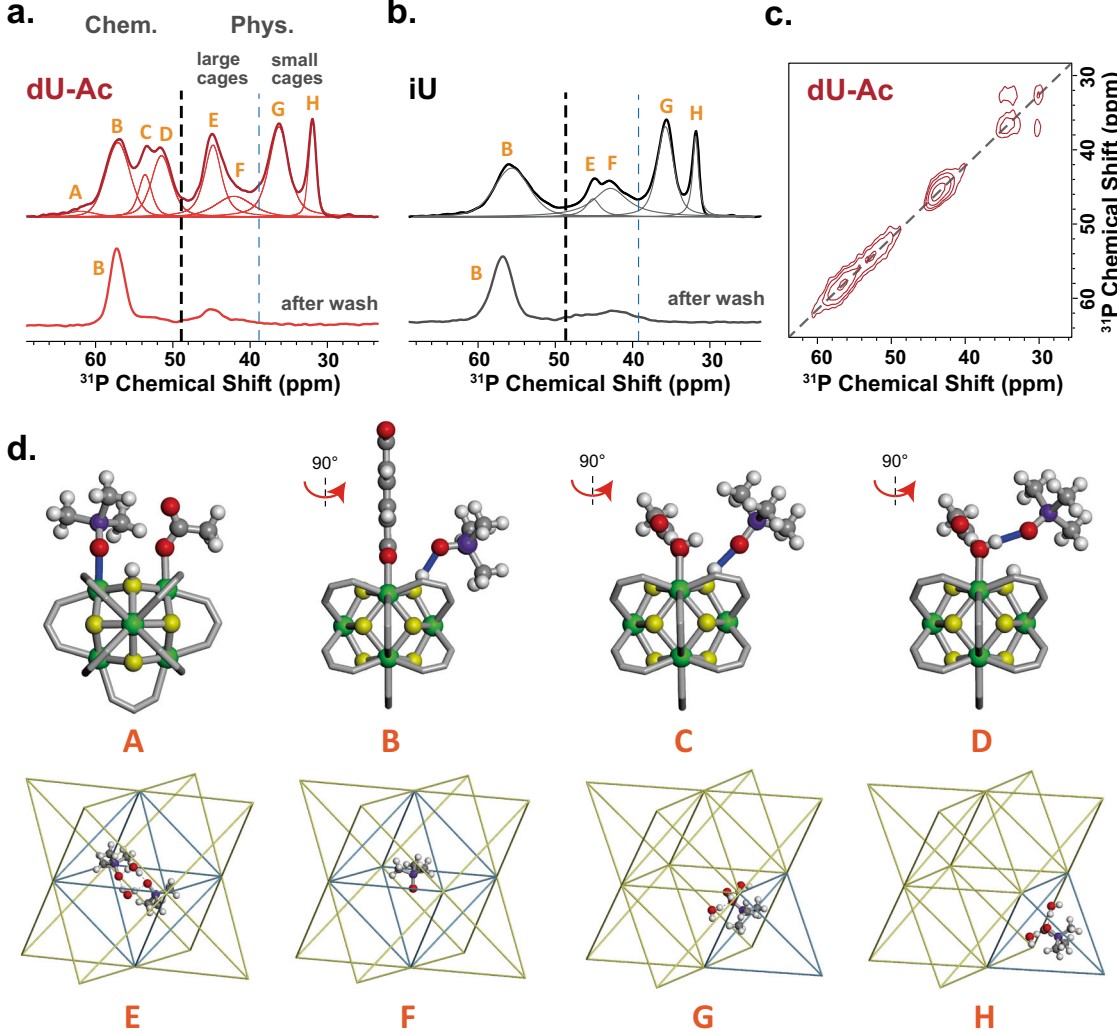

**Fig. 3 | Adsorption of TMPO.** [1]H-[31]P CP spectra of TMPO loaded in **a** dU-Ac (defective UiO-66 with acetate) and **b** iU (ideal UiO-66). The lower spectra are collected after washing with dichloromethane. The bold dotted lines separate the chemical shifts for physical and chemical adsorptions. The thin dotted lines separate the chemical shifts for physical adsorptions in large and small cages. The letters A-H correspond to different adsorption sites. **c** [31]P-[31]P 2D correlation spectrum of TMPO in dU-Ac. The mixing time is 10 ms and assisted with radio frequency driven recoupling (RFDR); **d** The configurations of adsorbed TMPO. The atoms are colored green for Zr, yellow for μ-O, red for O at defect sites and those of TMPO, gray for C, purple for P, and white for H.

limited part associated with defects or the outer surface. The quantifications of total adsorbed molecules (measured as the molar concentrations with respect to the BDC linkers) are shown in Fig. 4a, which is achieved by [1]H solution-state NMR of the digested samples (Supplementary Fig. 6).

The uptake of TMPO in defective UiO-66 (~0.82) is roughly twice of that in ideal UiO-66 (~0.42). It is estimated that 50% of the pores are defective in this dU sample. The uptake of TBPO in the defective UiO-66 is ~0.36 while its uptake in ideal UiO-66 is only ~0.033. We therefore conclude that most TBPO can only access defective pores. The uptake of TOPO is rather low (<0.01) in both ideal and defective UiO-66. The very few adsorbed TOPO molecules are most likely on the surface of the MOF particles (Fig. 4b).

The [31]P chemical shifts of TBPO and TOPO can also provide chemical distinction on sites they adsorb onto. Note that the overall signals of TBPO and TOPO shift about 8 ppm downfield as compared to TMPO[79]. The separation between chemically and physically adsorbed signals is marked with dotted lines (Figs. 4c, d). For TBPO, the chemically adsorbed species show a similar [31]P shift range as that of TMPO (except the 8 ppm offset). There is only one type of physically adsorbed TBPO in UiO-66, which should be attributed to the ones in the

large cages. For TOPO, its signals are related to the surface structure of UiO-66 particles which should be interesting for further investigations focused on the surface chemistry of MOFs.

**The quantification of defect concentration**

Besides revealing the details of local chemistry and pore geometry in defective MOFs, the [31]P signal of probe molecules can also quantify the relative fractions of different adsorption sites. Here, direct polarization (DP) of [31]P signals (Fig. 5a) is used to ensure reliable quantification results. The [31]P cross-polarization (CP) signals might not be quantitative as the CP efficiency and dynamics[81] could vary for different sites (Supplementary Fig. 7). We demonstrate the quantitative results by correlating the intensities of species C and D (Fig. 3d), i.e., the dominant defect-associated [31]P signals, to thermogravimetric analysis (TGA) measurements (Supplementary Fig. 8). The plot shows the general linear relation between the two very different measurements. Compared to the conventional TGA method[42,43], our [31]P NMR methods can provide much-needed molecular details of the defects.

In summary, we established a method based on [31]P NMR of probe molecules to characterize the internal chemistry in MOFs with ideal or defective lattices. The Brønsted and Lewis acidity of defects and

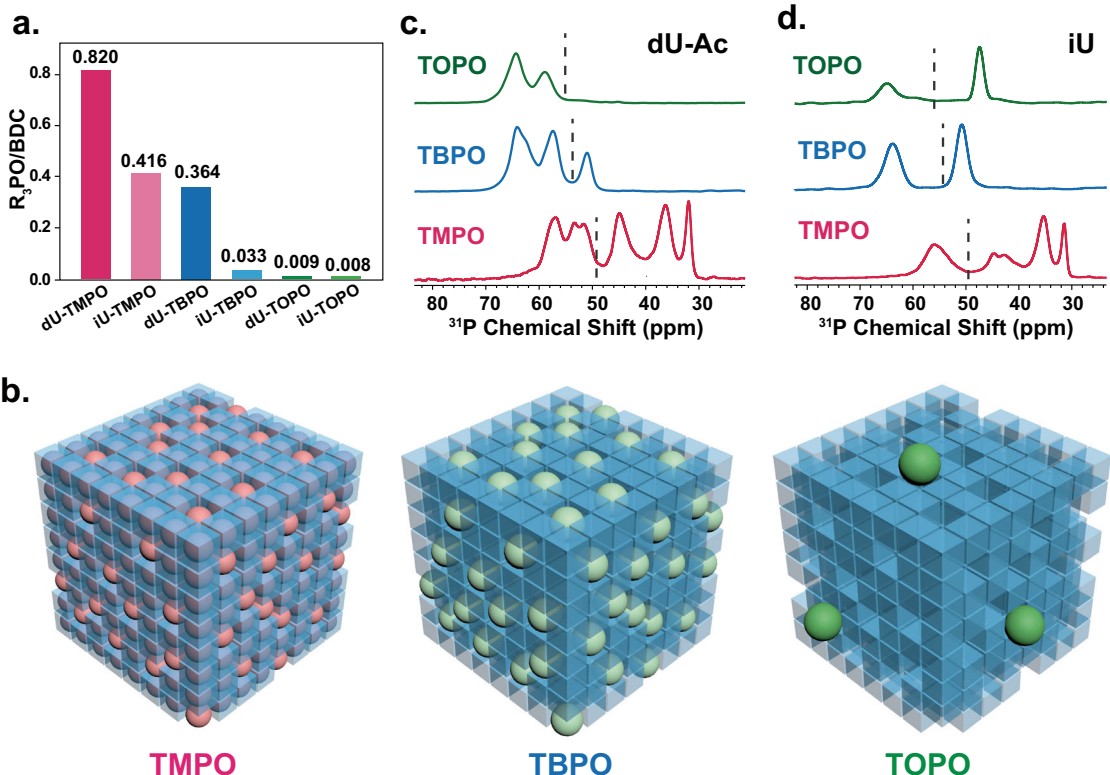

**Fig. 4 | Accessibility of defects. a** The molar concentrations of TMPO and its homologs loaded in UiO-66 as determined by [1]H dissolution NMR; **b** Schematic diagrams of the accessibility of different probe molecules in defective UiO-66. The spheres represent the probe molecules, the colored cubes represent the cages of non-defective lattice, and the empty cubes represent the defective pores. [31]P CP spectra of the probe molecules adsorbed in **c** dU-Ac and **d** iU. The dotted lines separate the chemical shifts for physical and chemical adsorption.

various bonding configurations of TMP and TMPO adsorption sites have been identified. The volumetric accessibility of pores and defects is revealed by using the homolog series of TMPO with different molecular sizes. The [31]P NMR offers both molecular details and reliable quantifications of defects and local environments in MOFs. This method can be adaptable for MOFs with various kinds of pore geometries, metal centers or functional groups. Further utilization of other probe molecules of different chemical affinity or molecular sizes is worthwhile for revealing defects of special chemistry and accessibility.

## Methods
### Synthesis procedures
**Synthesis of ideal UiO-66.** The preparation of ideal UiO-66 follows the procedure described in ref. 42. 5.39 g terephthalic acid (Energy Chemical, China), 3.78 g zirconium tetrachloride (Aladdin, China), and 2.86 mL hydrochloric acid (36-38 wt.%, SCR, China) were added in a flask containing 100 mL of N, N′-dimethylformamide (DMF) (SCR, China). The mixture was stirred at 70 °C to make the solids dissolve completely. The synthesis mixture was transferred to a 150 mL Teflon liner, then sealed in a stainless-steel autoclave. The autoclave was heated in the 220 °C oven for 24 h and UiO-66 particles were formed. The resulting solids were washed with DMF three times. Then, solids were soaked in methanol (SCR, China) for a day to exchange the solvent and the step was repeated three times. Finally, the solids were dried in a 70 °C vacuum oven for 48 h to remove the solvent and then ground into powder.

**Synthesis of defective UiO-66.** The preparation of defective UiO-66 follows the procedure described in ref. 25. 1.00 g terephthalic acid and 1.40 g zirconium tetrachloride were dissolved in 100 mL N, N′-dimethylformamide separately. The solutions were separated into different batches. In each batch, 10 mL of terephthalic acid solution, 10 mL of zirconium tetrachloride solution, and 1–6 mL acetic acid (SCR, China) were mixed in a 40 mL glass vial. The mixtures were placed in an 85 °C oven for 48 h, the washing and drying treatments of resulting UiO-66 particles were the same as above.

### Adsorption procedures
**Adsorption of TMPO and its homologs.** The adsorption steps follow the guidance described in ref. 66 with some minor modifications. Before adsorption, the UiO-66 powders were activated at 150 °C under vacuum for 4 h. 100 mg TMPO (or 240 mg TBPO, or 428 mg TOPO) (Alfa Aesar) was dissolved in 15 mL dichloromethane (Yonghua Chemical, China) under Ar atmosphere. 50 mg activated UiO-66 powders were added into the solution with 1 h of ultrasonic dispersion. Then the solids were centrifuged out and dried in a 70 °C vacuum oven for 24 h to remove the solvent.

**Adsorption with the additional washing step.** After UiO-66 sample was centrifuged out from TMPO solution, the solids were washed with 10 mL dichloromethane with ultrasonic dispersion. Then the solids were centrifuged out and dried in a 70 °C vacuum oven for 24 h to remove the solvent. Attention: the danger of using dichloromethane must be reminded.[82]

**Adsorption of TMP.** 50 mg activated UiO-66 sample was placed in a glass vial and was transferred into the nitrogen dry box. 0.3 mL TMP (Aldrich) was added into the vial and let it reach equilibrium for 24 h. The sample was vacuumed at room temperature to remove extra TMP.

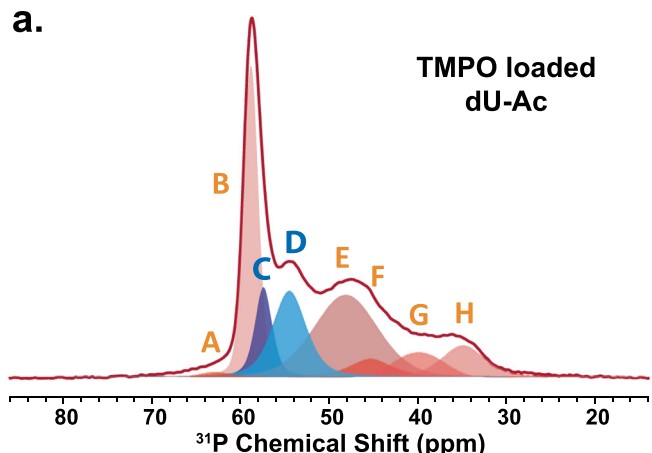

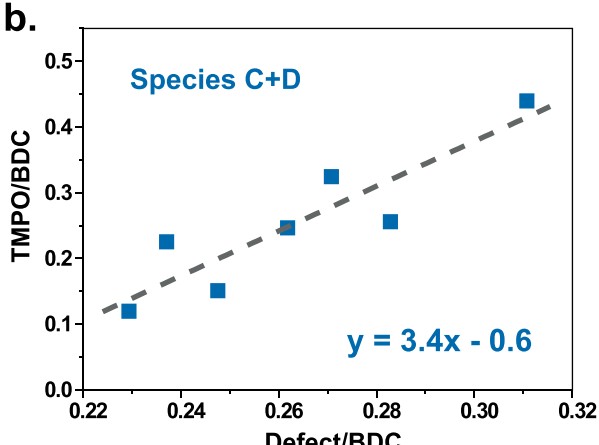

**Fig. 5 | Quantification of defects. a** A representative [31]P direct polarization (DP) spectrum of TMPO loaded in dU-Ac. Letters A-H correspond to different adsorption sites shown in Fig. 3d. The blue deconvoluted peaks represent the species C and D of TMPO which are associated with defects. **b** The molar concentrations of species C and D (measured by [31]P NMR) versus the defect concentrations measured by TGA. The dashed line corresponds to the fitting curve of the linear equation.

## Characterization methods

**Solution-state NMR**. [1]H solution-state NMR spectra were collected on a Bruker AVANCE III 400 MHz NMR spectrometer. The samples of interested were fully dissolved in 2 M NaOH solution (D$_2$O) and loaded in NMR tubes. The recycle delay was set to 6 s. The quantifications of probe molecules were determined by the integrals of their [1]H signals, normalized by the integral of the benzene [1]H signal of terephthalic acid.

**Solid-state NMR**. All the experiments were performed on a Bruker AVANCE III HD 600 MHz spectrometer at resonance frequencies of 150.84 MHz for [13]C, 242.80 MHz for [31]P, and 600.13 MHz for [1]H. Magic angle spinning (MAS) experiments were performed on 3.2 mm MAS probes at a spinning speed of 15 or 18 kHz. The [1]H signals were referenced to those of adamantane at 1.8 ppm. The [31]P signals were referenced to that of NH$_4$H$_2$PO$_4$ at 0.81 ppm.

For CPMAS, the [1]H-[31]P cross-polarization contact time was 4.0 ms and the [1]H decoupling power was 104 kHz. For [1]H-[31]P heteronuclear correlation (HETCOR), the CP contact time was 2.0 ms and the proton spin diffusion time was 0.1 ms. For [31]P-[31]P 2D correlation spectra, the CP contact time was 2.0 ms and the 10 ms mixing time was filled with radio frequency-driven recoupling (RFDR) pulses.

For [31]P direct polarization (DP) quantification experiments, the recycle delay was set to 15 s which is five times of the [31]P spin-lattice relaxation time (Supplementary Fig. 7). The [1]H decoupling power was

100 kHz. The DP spectra were deconvoluted into eight Gaussian peaks (species A to H) and their molar factions were determined by their integral areas. The total amount of adsorbed molecules was determined by [1]H solution-state NMR of digested samples.

**Powder X-ray diffraction (PXRD)**. The PXRD patterns were obtained on a Rigaku Ultima IV diffractometer using Cu-Kα radiation ($\lambda = 1.5418$ Å, tube operating at 40 kV and 30 mA), the 2θ range is 3°–50° and the scanning speed is 10°/min.

**Nitrogen sorption measurements**. Nitrogen sorption measurements were performed on a Belsorp max instrument at 77 K. The samples were pretreated under vacuum at 150 °C for 5 h. Approximately 40 mg of activated sample was used in each experiment.

**Transmission electron microscope (TEM)**. TEM images were obtained on a Hitachi HT7700 microscope.

**Thermogravimetric Analysis (TGA)**. TGA experiments were conducted on an SDT Q600 TGA instrument. The samples were heated from room temperature to 700 °C at a rate of 10 °C/min under 120.0 mL/min dry air flow.

The TGA curves can be divided into three parts according to the drops and plateaus: (1) from room temperature to 100 °C, the removal of solvents and adsorbed gases; (2) 100–350 °C, the loss of hydrogen bonding water, hydroxyl groups, and acetate modulator; (3) 350–700 °C, the loss of organic linkers (i.e., BDC) and the UiO-66 breaks down into zirconium oxide, ZrO$_2$.

The final mass of ZrO$_2$ was normalized to 100% (remaining weight at 600 °C). The weight of ideal UiO-66 at 350 °C is 220.2% which agrees with the chemical formula of Zr$_6$O$_6$(BDC)$_6$, In contrast, the weight of defective UiO-66 at 350 °C is below 220.2% suggesting the deficiency of BDC linkers. The defective UiO-66 takes the formula of Zr$_6$O$_{6+x}$(BDC)$_{6-x}$ where $x$ is the number of linker deficiency. $x$ can be calculated by the equation below.

$$\frac{w\%(350\,°C) - 100\%(600\,°C)}{220.2\% - 100\%} = \frac{6-x}{6}$$

here, w%(350 °C) is the normalized weight at 350 °C.
Defect concentration is calculated as

$$\text{defect concentration} = \frac{x}{6-x}$$

## Theoretical calculations

The details for theoretical calculations are described in the Supplementary Methods.

## Data availability

The datasets generated during and/or analyzed during the current study are available in the Supplementary Information or can be obtained from the corresponding author on request.

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

## Acknowledgements

This work was supported by the National Natural Science Foundation of China 21922410 (X.K.), 22072133 (X.K.), 21673206 (Q.W.); Zhejiang Provincial Natural Science Foundation LR19B050001 (X.K.); Leading Innovation and Entrepreneurship Team of Zhejiang Province 2020R01003 (X.K.).

## Author contributions

X.K. proposed and supervised the project. J.Y. performed the sample preparations and NMR characterizations, Z.K. and B.C. performed the computational studies. X.Y., W.C., W.S., Y.Z., A.Z., and Q.W. participated in the discussion. Y.F., W.C.C., H.G., and Y.Y., provided assistance for the experiments and data analysis.

## Competing interests

The authors declare no competing interests.
