## [Peer Review File · Nature Communications]

REVIEWER COMMENTS

Reviewer #1 (Remarks to the Author):

Dear Dr. Geddes,
manuscript NCOMMS-22-12144 reports on an exciting and highly topical aspect of research on metal-organic frameworks – the characterisation of local properties using probe molecules. This idea is well established and has been applied frequently in the past. E.g. preferred binding sites, unsaturated coordination sites, the strength of acidic binding sites and defects in MOFs were analysed very successfully. NMR and FTIR spectroscopy is frequently used as a detection method in this respect, as both allow to tune in on the properties of the probe molecules.

The authors introduce an alternate type of probe molecules based on phosphine and phosphine oxide derivatives. They contain a phosphorous core, which can be analysed with high sensitivity by ^{31}P NMR spectroscopy. The experiments on a series of UiO-66 samples with varying degrees of defects at the zirconium clusters demonstrate nicely that the chemical changes are pretty characteristic of potential binding scenarios. DFT calculations on cluster models substantiate this convincingly. By using single pulse excitation, even quantitative information about the proportions of the individual binding sites could be derived. Considering the expressiveness of the ^{31}P NMR spectroscopic shifts and that quantitative information can be obtained, I am convinced that the manuscript should be brought to the attention of a broad scientific community. Therefore, I recommend the manuscript for publication in Nature Communication.

Two aspects should be improved, however, before publication:
First, more details about the model clusters used for the DFT calculations should be included. Currently, no information besides the cut-outs in the figures is given. This is important as terminating the bdc molecules significantly influences the calculated chemical shifts. Also, in cases where only weak interactions between the probe molecules and the MOF occur, the setup of the framework around the probe molecules is of relevance.
Second, the manuscript neglects previous literature on studying the local properties of MOFs with NMR spectroscopy and probe molecules nearly completely. Considering this topic's well-developed state, I find this hardly acceptable. The authors should include an overview of previous findings from leading groups like the Reimer, Bertmer, Mafra, Senker, Mali and Brunner groups to name just a few.

Reviewer #2 (Remarks to the Author):

I have reviewed the manuscript entitled "molecular Identification and Quantification of Defect Sites in Metal-organic Frameworks with NMR Probe Molecules." by Kong and co-workers. The authors take a thorough approach to exploring defects in UiO-66 by using solid-state NMR and a series of phosphorus probe molecules. The work is very interesting and I really enjoyed reading the manuscript. Despite this, I don't feel that the work is exciting enough to warrant a Nature Communications result. I feel that this work is better suited for J. Phys. Chem. C or JPCC. The work is routine phosphorous NMR chemistry building off of known phosphorous NMR chemistry. I believe the work will be well cited in these other journals (I certainly would cite this work when I'm exploring the roll of defects). I do have a few suggestions prior to re-submission of this work to a more suitable journal (I don't believe addressing these reviews would warrant acceptance of this work in Nature Communications):

- a) the authors like to use the term under-coordinated zirconium, but the zirconium is not under-coordinated. It is simply coordinated by something other than BDC. Most likely water/hydroxide or acetate (or a mixture of both depending on the synthetic conditions).
- b) in general, the authors state that water bound and acetate bound defects but the text reads as if it is 100% one or 100% the other. Some more clarity should be put into the intro to put the reader into the same frame of mind as the authors. The work is very close to this but I occasionally found myself confused.
- c) electron-withdrawing not electron-drawing oxygen (although if oxygen could draw an electron, then I would be excited to see that art work).

These are all minor suggestions, and they are not a criticism of the work. Personally, I would recommend expanding the paper slightly and submitting it to a more suitable/specialised journal. I simply don't see the impact that I would expect from Nature Communications here.

Reviewer #3 (Remarks to the Author):

This manuscript described using phosphine (TMP) and phosphine oxide (TMPO, TBPO and TPPO) based probe molecules to detect the defects in MOFs. Defect engineering is one of the most effective approaches to tune the properties of MOFs for targeted applications. Characterizing defects is very challenging. To this end, this work is very timely and important. The novelty of this work lies in the clever use of the above probe molecules, despite that they have been widely used for characterization of the acidity of microporous and mesoporous materials. The ability to quantifying the defect concentration adds additional value to the work. The experiments appear to be carried out carefully and data interpreted correctly. The results should be of interest to MOF community. I recommend the paper be accepted upon some minor revisions.

1. TEM images are not unambiguous to confirm the formation of defects. The authors should independently use other method to validate their NMR method.
2. The structures of probe molecules in Fig. 1c should be redrawn following the normal protocol to show 3D structure.
3. The referencing could have been better! For example, a very recent paper involving direct characterization of MOF defect by multinuclear SSNMR should be cited:
<https://doi.org/10.1016/j.ssnmr.2022.101793>
4. Some probe molecules such as TMP are very sensitive to air and trace amount of water. The authors should emphasize what precaution was taken.

Point-by-point response

Reviewer #1 (Remarks to the Author):

Manuscript NCOMMS-22-12144 reports on an exciting and highly topical aspect of research on metal-organic frameworks – the characterisation of local properties using probe molecules. This idea is well established and has been applied frequently in the past. E.g. preferred binding sites, unsaturated coordination sites, the strength of acidic binding sites and defects in MOFs were analysed very successfully. NMR and FTIR spectroscopy is frequently used as a detection method in this respect, as both allow to tune in on the properties of the probe molecules.

The authors introduce an alternate type of probe molecules based on phosphine and phosphine oxide derivatives. They contain a phosphorous core, which can be analysed with high sensitivity by ^{31}P NMR spectroscopy. The experiments on a series of UiO-66 samples with varying degrees of defects at the zirconium clusters demonstrate nicely that the chemical changes are pretty characteristic of potential binding scenarios. DFT calculations on cluster models substantiate this convincingly. By using single pulse excitation, even quantitative information about the proportions of the individual binding sites could be derived. Considering the expressiveness of the ^{31}P NMR spectroscopic shifts and that quantitative information can be obtained, I am convinced that the manuscript should be brought to the attention of a broad scientific community. Therefore, I recommend the manuscript for publication in Nature Communication.

Two aspects should be improved, however, before publication:

Q: First, more details about the model clusters used for the DFT calculations should be included. Currently, no information besides the cut-outs in the figures is given. This is important as terminating the bdc molecules significantly influences the calculated chemical shifts. Also, in cases where only weak interactions between the probe molecules and the MOF occur, the setup of the framework around the probe molecules is of relevance.

A: Thank you for the advice. We now upload the cif files for all model clusters for DFT calculations. We also add a full paragraph in the revised supporting information to explain the key considerations in our simulation.

Q: Second, the manuscript neglects previous literature on studying the local properties of MOFs with NMR spectroscopy and probe molecules nearly completely. Considering this topic's well-developed state, I find this hardly acceptable. The authors should include an overview of previous findings from leading groups like the Reimer, Bertmer, Mafra, Senker, Mali and Brunner groups to name just a few.

A: Thanks for the criticism. We did miss a few references from the leading research groups. We modify the introduction to accommodate an overview of their important findings.

Reviewer #2 (Remarks to the Author):

I have reviewed the manuscript entitled "molecular Identification and Quantification of Defect Sites in Metal-organic Frameworks with NMR Probe Molecules." by Kong and co-workers. The authors take a through approach to exploring defects in UiO-66 by using solid-state NMR and a series of phosphorus probe molecules. The work is very interesting and I really enjoyed reading the manuscript. Despite this, I don't feel that the work is exciting enough to warrant a Nature Communications result. I feel that this work is better suited for J. Phys. Chem. C or JPCC. The work is routine phosphorous NMR chemistry building off of known phosphorous NMR chemistry. I believe the work will be well cited in these other journals (I certainly would cite this work when I'm exploring the roll of defects). I do have a few suggestions prior to re-submission of this work to a more suitable journal (I don't believe addressing these reviews would warrant acceptance of this work in Nature Communications):

Q: a) the authors like to use the term under-coordinated zirconium, but the zirconium is not under-coordinated. It is simply coordinated by something other than BDC. Most likely water/hydroxide or acetate (or a mixture of both depending on the synthetic conditions).

A: We use the term "under-coordinated zirconium" to describe defect zirconium without compensating acetate, water or hydroxide. Only the such under-coordinated zirconium can form Lewis acid sites in defective MOFs. We further explain the terminology in the revised manuscript: "In a defective MOF, some linkers are missing on defect zirconium atoms. Normally, most of the defect zirconium are compensated by acetate modulators and water (in dU-Ac) creating new Brønsted acid sites. Only a few modulators could fall off, leaving under-coordinated zirconium as Lewis acid sites. In contrast, the under-coordinated zirconium sites could be abundant in structures lack of acetate (in dU-no Ac)."

Q: b) in general, the authors state that water bound and acetate bound defects but the text reads as if it is 100% one or 100% the other. Some more clarity should be put into the intro to put the reader into the same frame of mind as the authors. The work is very close to this but I occasionally found myself confused.

A: Thanks for the advice. We modify the sentence to make the description clearer: "Here, the Brønsted site is formed by the coexistence of water and acetate." We also made a slight revision in the intro to avoid any misleading statement.

Q: c) electron-withdrawing not electron-drawing oxygen (although if oxygen could draw an electron, then I would be excited to see that art work).

A: Thank you for pointing out the typo. It is now corrected.

These are all minor suggestions, and they are not a criticism of the work. Personally, I would recommend expanding the paper slightly and submitting it to a more suitable/specialised journal. I simply don't see the impact that I would expect from Nature Communications here.

A: Here we use our self-evaluation in the cover letter to address the reviewer's concern: "Our work is the first demonstration of the ^{31}P NMR strategy in studying the defects in MOFs. Conventionally, the ^{31}P NMR strategy has been utilized to characterize acidity in crystalline solids (Chem. Rev. 2017, 117, 12475–12531). But the defects in MOFs do not have well-defined local chemical environments as those ideal crystalline matrices. It took tremendous effort to design the measurement procedure, to derive the details of the coordination structures, and to achieve quantitative results. In a previous work, Prof. Yaghi's group has used a similar ^{31}P NMR method to measure acidity in MOF (Nat. Chem. 2019, 11, 170–176). However, that work was only limited to non-defective MOFs, and it provided no quantitative information. Our study is a large step forward."

Reviewer #3 (Remarks to the Author):

This manuscript described using phosphine (TMP) and phosphine oxide (TMPO, TBPO and TPPO) based probe molecules to detect the defects in MOFs. Defect engineering is one of the most effective approaches to tune the properties of MOFs for targeted applications. Characterizing defects is very challenging. To this end, this work is very timely and important. The novelty of this work lies in the clever use of the above probe molecules, despite that they have been widely used for characterization of the acidity of microporous and mesoporous materials. The ability to quantifying the defect concentration adds additional value to the work. The experiments appear to be carried out carefully and data interpreted correctly. The results should be of interest to MOF community. I recommend the paper be accepted upon some minor revisions.

Q: 1. TEM images are not unambiguous to confirm the formation of defects. The authors should independently use other method to validate their NMR method.

A: In fact, we used TGA to confirm the formation of defects. We add a sentence to make this clear: "The formation of defects was determined by the thermogravimetric analysis (TGA) (see supporting information)".

Q: 2. The structures of probe molecules in Fig. 1c should be redrawn following the normal protocol to show 3D structure.

A: Thanks for the advice, and we redraw Fig. 1c to show the 3D structure.

Q: 3. The referencing could have been better! For example, a very recent paper involving direct characterization of MOF defect by multinuclear SSNMR should be cited: <https://doi.org/10.1016/j.ssnmr.2022.101793>

A: Thank you for the wonderful suggestion. The reference is now cited.

Q: 4. Some probe molecules such as TMP are very sensitive to air and trace amount of water. The authors should emphasize what precaution was taken.

A: Thank you for the notice. We add the precaution that "Note that TMP is easily oxidized into

TMPO, while TMPO, TBPO and TOPO are sensitive to moisture. The samples were prepared in Ar or N₂ glove boxes to avoid the environmental influence.”

REVIEWERS' COMMENTS:

Reviewer #1 (Remarks to the Author):

I have reviewed the updated version of the manuscript NCOMMS-22-12144A. The authors have considered the reviewers' comments, and I have no further reservations about its publication in Nature Communication.

Reviewer #3 (Remarks to the Author):

The authors have addressed my concerns. I recommend it be accepted for publication.